# Targeting Mitochondria by SS-31 Ameliorates the Whole Body Energy Status in Cancer- and Chemotherapy-Induced Cachexia

**DOI:** 10.3390/cancers13040850

**Published:** 2021-02-18

**Authors:** Riccardo Ballarò, Patrizia Lopalco, Valentina Audrito, Marc Beltrà, Fabrizio Pin, Roberto Angelini, Paola Costelli, Angela Corcelli, Andrea Bonetto, Hazel H. Szeto, Thomas M. O’Connell, Fabio Penna

**Affiliations:** 1Department of Clinical and Biological Sciences, University of Torino, 10125 Torino, Italy; RBallaro@mdanderson.org (R.B.); marc.beltrabach@unito.it (M.B.); paola.costelli@unito.it (P.C.); 2Interuniversity Institute of Myology, 61029 Urbino, Italy; 3Dipartimento di Scienze Mediche di Base, Neuroscienze e Organi di Senso, University of Bari Aldo Moro, 70121 Bari, Italy; patrizia.lopalco@uniba.it (P.L.); angela.corcelli@uniba.it (A.C.); 4Molecular Biotechnology Center, University of Torino, 10125 Torino, Italy; valentina.audrito@unito.it; 5Department of Anatomy, Cell Biology and Physiology, Indiana University, Indianapolis, IN 46202, USA; fpin@iu.edu; 6Swansea University Medical School, Swansea University, Swansea SA2 8PP, UK; roberto.angelini@swansea.ac.uk; 7Department of Surgery, Indiana University, Indianapolis, IN 46202, USA; abonetto@iu.edu; 8Social Profit Network Research Lab, New York, NY 10016, USA; hhszeto@med.cornell.edu; 9Department of Otolaryngology, Indiana University School of Medicine, Indianapolis, IN 46202, USA; thoconne@iu.edu

**Keywords:** cancer cachexia, muscle wasting, mitochondria, SS-31, metabolomics, liver

## Abstract

**Simple Summary:**

Cancer cachexia is a debilitating syndrome, caused by both tumor growth and chemotherapy. The skeletal muscle is one of the main tissues affected during cachexia, presenting with altered metabolism and function, leading to progressive tissue wasting. In the current study we aimed at counteracting cachexia by pharmacologically improving metabolic function with the mitochondria-targeted compound SS-31. Experimental cancer cachexia was obtained using C26-bearing mice either receiving chemotherapy (oxaliplatin plus 5-fluorouracil) or not. SS-31 proved effective in rescuing some of the metabolic impairments imposed by both tumor and chemotherapy in the skeletal muscle and the liver, improving systemic energy control. Unfortunately, such effects were no longer present at late disease stages when refractory cachexia ensued. Overall, we provide evidence of potential new treatments targeting mitochondrial function in order to counteract or delay cancer cachexia.

**Abstract:**

*Objective*: Cachexia is a complex metabolic syndrome frequently occurring in cancer patients and exacerbated by chemotherapy. In skeletal muscle of cancer hosts, reduced oxidative capacity and low intracellular ATP resulting from abnormal mitochondrial function were described. *Methods*: The present study aimed at evaluating the ability of the mitochondria-targeted compound SS-31 to counteract muscle wasting and altered metabolism in C26-bearing (C26) mice either receiving chemotherapy (OXFU: oxaliplatin plus 5-fluorouracil) or not. *Results*: Mitochondrial dysfunction in C26-bearing (C26) mice associated with alterations of cardiolipin fatty acid chains. Selectively targeting cardiolipin with SS-31 partially counteracted body wasting and prevented the reduction of glycolytic myofiber area. SS-31 prompted muscle mitochondrial succinate dehydrogenase (SDH) activity and rescued intracellular ATP levels, although it was unable to counteract mitochondrial protein loss. Progressively increased dosing of SS-31 to C26 OXFU mice showed transient (21 days) beneficial effects on body and muscle weight loss before the onset of a refractory end-stage condition (28 days). At day 21, SS-31 prevented mitochondrial loss and abnormal autophagy/mitophagy. Skeletal muscle, liver and plasma metabolomes were analyzed, showing marked energy and protein metabolism alterations in tumor hosts. SS-31 partially modulated skeletal muscle and liver metabolome, likely reflecting an improved systemic energy homeostasis. *Conclusions*: The results suggest that targeting mitochondrial function may be as important as targeting protein anabolism/catabolism for the prevention of cancer cachexia. With this in mind, prospective multi-modal therapies including SS-31 are warranted.

## 1. Introduction

Cachexia, a complex disorder frequently associated with cancer, is a wasting syndrome characterized by hypercatabolism in both the skeletal muscle and the adipose tissue that significantly complicates patient management and reduces the tolerance and the effectiveness of anti-cancer treatments, eventually resulting in cancer patient death in up to 30% of the cases [1]. Despite the relevance of cancer cachexia to patient quality of life and survival, effective treatments are still lacking. So far, most of the efforts aimed at counteracting cachexia are focused on skeletal muscle sparing, in order to maintain the patients in good physical shape, allowing to adopt the most effective as possible anti-cancer treatment protocol. 

Muscle protein hypercatabolism has been proposed to result from the direct action of tumor-derived signaling factors and host-driven inflammatory response [2]. Conversely, since cachexia is a systemic disorder that includes both anorexia and increased resting energy expenditure, the altered energy metabolism at both muscle and whole body levels may be the main indirect cause of tissue hypercatabolism [3], potentially representing an ideal target for future interventions. Besides the imbalance between energy intake and consumption, the energy deficit in cachexia is likely a consequence of tissue metabolic dysfunctions. Such hypothesis is supported by several reports showing skeletal muscle mitochondrial alterations in tumor hosts (reviewed in [4]). From the metabolic point of view, reduced oxidative capacity and low intracellular ATP content have been found in the skeletal muscle of cachectic animals [5]. Beyond tumor-driven alterations, chemotherapy also negatively affects muscle mass and function strongly impacting on muscle and systemic energy metabolism [6]. We recently reported that, at least in experimental cancer cachexia, muscle mitochondrial dysfunction is linked to protein hypercatabolism, excessive autophagy and mitophagy being associated with impaired mitochondrial respiration [7]. The experimental demonstration in tumor-bearing mice that mitochondrial alterations precede muscle hypercatabolism and mass loss [8] prompts the optimization of prospective therapeutic approaches targeting mitochondria. On this line, several compounds aimed at improving the energetic status by acting through distinct mechanisms and mainly indirectly targeting mitochondria have been recently tested in experimental cancer cachexia and are in support of this hypothesis (reviewed in [4,9]). Despite the partial beneficial action of such molecules, none of them directly improved mitochondrial efficiency. The Szeto-Schiller peptide (SS-31) proved effective in counteracting mitochondrial dysfunction observed in aging and in several diseases by targeting cardiolipin [10,11,12,13,14,15,16], a phospholipid essential for cristae structure and overall mitochondrial function [17,18]. 

The present study aimed at testing whether boosting mitochondrial function by means of systemic SS-31 administration could benefit muscle wasting in tumor-bearing mice, either untreated of chronically exposed to chemotherapy, by improving mitochondrial function and affecting the whole body metabolome. The results suggest that SS-31 exerts beneficial actions in tumor-bearing animals, irrespective of chemotherapy exposure. Such effects appear mainly achieved by acting on whole body metabolism rather than targeting muscle specific molecular alterations. 

## 2. Results

### 2.1. Targeting Cardiolipin with SS-31 Partially Ameliorates Cachexia without Interfering with Tumor Growth

Preliminarily observations show alterations in cardiolipin and phosphatidylglycerol (a cardiolipin precursor) fatty acid chains, namely the reduction of the species bearing oleic acid (18:1) chains, in mitochondria isolated from gastrocnemius of C26 hosts as compared to controls animals (Appendix A), supporting the idea that tumor growth triggers an intrinsic alteration of muscle mitochondria. 

In order to target muscle mitochondria, SS-31 was administered daily to C26-bearing (C26) mice or to chemotherapy (OXFU)-treated C26 mice (see Appendix A for the experimental design), with the aim of understanding the effects of the drug in an experimental setting close to cancer patients receiving chemotherapy.

SS-31 was first tested in C26 cell cultures, alone or in combination with oxaliplatin or 5-fluorouracil to exclude any interference with chemotherapy-induced cytotoxicity. Cell proliferation, apoptosis as well as cell cycle distribution were not affected by SS-31 treatment (Appendix A).

As expected, oxaliplatin and 5-fluorouracil reduced cell proliferation, triggered apoptosis and cell cycle arrest in C26 cells (G_2_/M and S, respectively for oxaliplatin and 5-fluorouracil), irrespective of the exposure to SS-31 (Appendix A).

Moving to the in vivo study, growth of C26 tumors in mice for 14 days caused a 26% body weight loss compared to controls (Figure 1a), associated with lower mass of both tibialis anterior (TA; −36%; Figure 1b) and gastrocnemius (GSN; −36%; Figure 1c). As previously characterized [5], OXFU administration in C26 hosts (C26 OXFU) prolonged survival (100% survival until day 28 after tumor implantation), despite inducing severe body (−23% vs. Control; Figure 1a) and muscle weight loss (TA: −51%; GSN: −44%; Figure 1b,c). SS-31 administration (2 mg/kg) partly relieved body weight loss in C26-bearing mice (+ 11% vs. untreated C26; Figure 1a), whereas it did not correct muscle wasting (Figure 1b,c). Body and muscle weight loss were not prevented by SS-31 in C26 OXFU mice (Figure 1a–c). In control mice, muscle strength increased during the experimental period, and such increase was enhanced by SS-31 administration at early stages (Figure 1d). C26-induced muscle wasting was associated with loss of muscle strength in both C26 and C26 OXFU mice. Similarly to control mice, SS-31 improved muscle strength only after 14 days in C26 OXFU mice, but was ineffective in C26 mice (Figure 1d). Consistent with impaired muscle mass and strength, glycolytic and oxidative muscle fiber cross-sectional area (CSA) was lower in both C26 and C26 OXFU mice (Figure 1e). Of note, SS-31 preserved the size of glycolytic fibers in C26 hosts. On the other hand, SS-31 did not improve glycolytic or oxidative fiber CSA in C26 OXFU mice (Figure 1e).

Similarly to muscle mass, the gonadal white adipose tissue (WAT) weight was markedly less in C26 hosts (−94% vs. Control) and almost undetectable in C26 OXFU mice (Appendix A). SS-31 induced greater WAT weight in control mice (+25%) and a similar trend was observed also in C26 hosts (+9%; Appendix A). As for the other tissues, the heart weight was not affected by tumor growth, while liver mass was 32% greater than control and was not rescued by SS-31 administration (Appendix A). In line with our previous findings [5], tumor became resistant to OXFU administration, whereas, similar to our in vitro experiments (Appendix A), SS-31 did not impinge on tumor size (Appendix A).

### 2.2. SS-31 Improves Muscle Energetics, Not Mitochondrial Quantity

In order to understand the effects of SS-31 on muscle mitochondrial status, succinate dehydrogenase (SDH) activity, ATP levels and markers of mitochondrial biogenesis and quantity were assessed in GSN muscle of C26 mice, either untreated or treated with chemotherapy. Both C26 and C26 OXFU mice showed impaired SDH activity, the latter also showing decreased intracellular ATP levels (Figure 2a,b). SDH activity was rescued to control levels by SS-31 in both C26 and C26 OXFU mice, although the latter were not statistically different from the untreated C26 OXFU (Figure 2a). SS-31 administration did not significantly impact on ATP content (Figure 2b). The functional mitochondrial alterations in both C26 and C26 OXU mice associated with molecular changes, including reduced levels of the master regulator of mitochondrial biogenesis, PGC-1α, and decreased SDHA and cytochrome *c* content, both markers of mitochondrial abundance (Figure 2c). SS-31 was unable to prevent any of these alterations, pointing to a mechanism of action based on improved mitochondrial function rather than on the transcriptional control of mitochondria abundance (Figure 2c). 

Consistently, mitochondrial dysfunctions and negative energy balance associate with reduced protein anabolism, as confirmed by the severely impaired protein synthesis in C26-bearing mice (Appendix A). SS-31, although being able to restore SDH activity, was unable to significantly rescue protein anabolism.

### 2.3. Increasing SS-31 Dosage Is More Effective at Intermediate Time Points before Refractory Cachexia Onset

The partially encouraging effects of SS-31 in improving cancer-induced cachexia especially in short experimental settings (i.e., 14 days in untreated C26 hosts or intermediate time points upon OXFU treatment), as compared to the late refractory condition in C26 OXFU mice at 28 days, prompted us to investigate the effects of progressively increased SS-31 administration (2–5–10 mg/kg, weekly; Appendix A) in C26 OXFU mice also collecting tissues at intermediate time points (i.e., day 14 and 21 after tumor injection). The expected progressive loss of body weight in the C26 OXFU mice (−40% vs. Control) was relieved by SS-31 (+16% vs. untreated) at day 21 after tumor implantation (Figure 3a). A similar protection was observed in TA muscle (+21% vs. untreated; Figure 3b), while the drug was not able to exert any significant effect on the gastrocnemius (Figure 3c). Muscle function progressively declined in C26 OXFU mice as compared to control and baseline (Figure 3d). Muscle strength in SS-31-treated C26 OXFU animals was comparable to day 0 and significantly higher than in untreated tumor hosts at 14 and 28 days after tumor injection, respectively (Figure 3d). 

Body weight loss and muscle wasting in the C26 OXFU mice were paralleled by a progressive loss of WAT weight, that at at days 21 and 28 after tumor injection was totally depleted, irrespective of treatment with SS-31 (Appendix A). Heart weight was lower at days 14 and 28 in C26 OXFU mice whereas liver mass was not affected (Appendix A). The loss of heart mass observed in C26 OXFU mice at day 28 could not be appreciated in SS-31-treated mice (Appendix A). Furthermore, tumor size progressively increased at the different experimental points, irrespective of SS-31 administration (Appendix A).

In order to understand if the SS-31-induced improvements of body and muscle wasting were triggered by changes in mitochondrial homeostasis, the levels of mitochondrial proteins and autophagy/mitophagy markers were evaluated (Figure 4). In muscle mitochondria from C26 OXFU mice, cytochrome *c* decreased, whereas COX IV was not significantly affected (Figure 4a). Cytochrome *c* reduction was associated with increased autophagy-related protein LC3 II and increased mitophagy markers BNIP3 (mitochondria-localized BNIP3 homodimer; 60 kDa), PINK1 and parkin, thereby suggesting that the mitochondrial degradation machinery was overactivated in comparison to control animals (Figure 4a). LC3 II and BNIP3 (cytosolic monomer; 30 KDa) levels increased in the cytosolic fraction as well, while PGC-1α levels decreased (Figure 4b). SS-31 partially prevented LC3 II and PINK1 increase in the mitochondrial-enriched fraction (Figure 4a). Moreover, the altered PGC-1α levels observed in C26 OXFU mice were not detected in SS-31-treated animals (Figure 4b). The above reported molecular alterations in the muscle of the C26 OXFU mice resulted in a severe decrease of both ADP- and succinate-coupled ex vivo mitochondrial maximal respiration. Such impairment was not corrected by SS-31, suggesting that, despite counteracting some of the above-mentioned alterations, SS-31 cannot afford an effective restoration of ex vivo muscle mitochondrial respiration (Figure 4c).

The same mitochondrial analysis in the muscle of C26 OXFU mice was also performed at a terminal time point (day 28 after tumor injection; Appendix A). In the mitochondrial-enriched fraction, both COX IV and cytochrome *c* levels decreased, whereas LC3 II, PINK1 and parkin increased (Appendix A). In the cytosolic fraction, PGC-1α decrease was paralleled by LC3 II and BNIP3 accumulation (Appendix A). Similarly to day 21, mitochondrial protein and autophagy/mitophagy marker alterations were associated with a drop in mitochondrial respiratory capacity (Appendix A). Consistently with the notion of refractory cachexia, SS-31 administration did not significantly affect mitochondrial respiration. As for molecules pertaining to mitochondrial homeostasis, SS-31 was able to prevent parkin accumulation only (Appendix A).

### 2.4. Muscle, Liver and Plasma Metabolomes Reveal SS-31 Improvement of Systemic Energy Status

In order to understand how targeting mitochondria with SS-31 may affect local and systemic energy metabolism in cancer- and chemotherapy-induced cachexia, we analyzed the metabolomic profile of skeletal muscle, liver and plasma in C26 OXFU mice at day 21 after tumor injection using an NMR-based approach, as shown in [6]. Consistent with the marked changes in muscle mitochondrial function, several metabolites were severely affected (Figure 5a). A decrease in muscle NADH was observed in C26 OXFU mice (*p*-value = 0.051), supporting the emerging concept of cancer-induced perturbations to redox metabolism [19], also consistent with the reduced succinate levels. Increases in the amino acids glutamine, isoleucine, leucine, valine and phenylalanine were observed in C26 OXFU mice, suggesting increased protein degradation. None of these metabolomic changes were impacted by SS-31. Interestingly, the metabolites, alanine and ATP were the only ones that were clearly impacted by SS-31 treatment, which yielded a significant increase in both above the C26 OXFU group.

The liver metabolome demonstrated the greatest number of metabolic differences between the groups (Figure 5b). Boxplots of some of these significant differences are shown in Appendix A.

The glucose levels were significantly decreased in the C26 OXFU group and treatment with SS-31 yielded a significant recovery toward control levels. To further probe carbohydrate metabolism in the liver, the glycogen levels were measured and showed an even more dramatic decrease in the C26 OXFU group and a highly significant recovery with SS-31. Consistent with the muscle, the branched chain amino acids increased in the C26 OXFU group and were not significantly impacted by SS-31. The liver glutathione levels were depleted in the C26 OXFU group and treatment with SS-31 led to a significant recovery, suggesting an effect on the liver response to oxidative stress. The choline levels were significantly increased in C26 OXFU group, potentially indicating a dysregulation of fatty acid metabolism that was mitigated by SS-31. 

Significant alterations in the plasma metabolome were also observed (see Appendix A). Consistent with the liver results, glucose levels were significantly decreased and partially mitigated by SS-31. Reductions in amino acids including asparagine, glutamine, glycine methionine, threonine, tryptophan and tyrosine were observed in the C26 OXFU. Among the amino acids only glutamine was altered by SS-31 yielding recovery back to control levels.

## 3. Discussion

The present study aimed at testing the effectiveness of boosting mitochondrial function to counteract the metabolic changes induced by C26 tumor growth and chemotherapy, in the final attempt to improve cachexia. The results, integrating data at both tissue and systemic levels, either in untreated or SS-31 treated animals, highlight that cancer cachexia should be considered as a systemic energy wasting disorder [20]. Along this line, the current diagnostic criteria mainly based on body weight loss and inflammation are not considering circulating markers of metabolic alterations. Energy metabolism impairment can be easily detected non-invasively and may potentially allow to monitor the onset and progression of cachexia from a systemic point of view, especially with the aim of monitoring the response to anti-cachexia treatments.

So far, tumor-derived factors and systemic inflammation have been considered the main drivers of cachexia [2], although cytokine targeting or their use as biomarkers failed due to the high heterogeneity observed in cancer patients. The results reported in the present study push on mitochondrial efficiency as a tool to improve muscle and systemic energy metabolism. Along this line, the present dataset point to the energy homeostasis as a possible therapeutic target, also useful for monitoring treatment effectiveness and for predicting survival. Indeed, considering: a) the link between the loss of muscle mass and function and morbidity/mortality in cancer cachexia [21] and b) the relevance of the skeletal muscle to sustain whole body energy homeostasis [22], the causal association of energy failure with patient death is conceivable.

SS-31 effectiveness in the treatment of disorders associated with mitochondrial dysfunction has been ascribed to its ability to either boosting oxidative metabolism or counteracting oxidative stress [18]. In experimental cancer- and chemotherapy-induced cachexia, the observed mitochondrial alterations leading to impaired substrate oxidation are likely to rely on both intrinsic and extrinsic mechanisms. As for the intrinsic ones, phosphatidylglycerol and cardiolipin alterations partly explains the reduced electron transport chain (SDH) activity and oxygen consumption. Mitochondrial phospholipid alterations are in line with previous data reporting significant compositional changes in cardiolipin chains occurring in cancer cells [23]. It was previously suggested that when cells rely only on fatty acid de novo biosynthesis, cardiolipin pools are mainly composed of palmitoleic and oleic acid [24]. The reduction of oleic acid chains in the phospholipids (See Appendix A), likely due to impaired fatty acid de novo biosynthesis, supports the idea that the metabolic shifts observed in tumor-bearing mice is triggered by a modification of muscle mitochondrial membranes having functional consequences. The rescue of SDH activity exerted by SS-31 is in line with its proposed ‘mitoprotective’ action mediated by the binding to cardiolipin-enriched membranes [25]. Indeed, SDH function was previously demonstrated to be cardiolipin dependent [26] and a recent study has shown that by binding to cardiolipin, SS-31 interacts with 12 proteins involved in ATP production [27]. Moreover, in experimental hearth failure, long-term SS-31 treatment counteracts disease progression even restoring cardiolipin content and mitochondrial dynamics [14,28]. By contrast, SS-31 inability to improve respirometric measures might suggest that mitochondrial damage is severe enough to exceed the potential benefits produced by the drug, although a deeper analysis of mitochondrial dynamics and ultrastructure is needed to clarify this point. Furthermore, poor drug pharmacokinetic in late-stage cachectic tumor-bearing and chemotherapy-treated mice, as already shown in cancer patients for other drug classes, potentially limiting the distribution of SS-31, should be taken into account [29]. Consistently, preliminary results (not shown) obtained adding SS-31 directly to isolated muscle mitochondria from both healthy and cachectic mice show increased oxygen consumption, suggesting that the mitochondria are still potentially benefiting from SS-31 treatment.

In tumor-bearing mice receiving chemotherapy, the progressive reduction of mitochondrial proteins involved in oxidative metabolism triggered by both increased mitophagy and impaired biogenesis is compatible with the limited effectiveness of SS-31, becoming ineffective in the more severe stages of cachexia where such alterations are exacerbated [5]. These observations suggest that SS-31 is likely able to boost mitochondrial efficiency in moderately damaged mitochondria, while it cannot promote mitochondrial biogenesis and only partially counteracts the excessive disposal through mitophagy. Increased mitophagy could result from intrinsic mitochondrial damage or from extrinsic phenomena such as enhancement of the autophagy flux and of mitophagy gene induction. Both extrinsic events occur in the skeletal muscle of tumor-bearing mice either receiving chemotherapy or not [5,7,30]. The limited SS-31 effects observed in C26 OXFU mice are in line with the ineffectiveness of skeletal muscle PGC-1α overexpression in the same severe experimental conditions [5]. Although in a different background, PGC-1α overexpression proved effective in counteracting muscle wasting in Lewis lung-bearing mice, a milder and slower model of cancer cachexia [31]. 

The improvement of mitochondrial function is the main mechanism underlying the effectiveness of endurance exercise in counteracting the loss of muscle mass and function in cancer cachexia, and the above-mentioned mitochondria-targeting approaches can be considered as exercise mimetics [32]. Considering that exercise feasibility in oncology patients is limited, because of several issues related to the cancer itself or to other comorbidities, rendering the patient not eligible or even exercise intolerant [33], SS-31, mimicking some of the beneficial effects of exercise, could widen the target audience for exercise-based interventions by both promoting exercise tolerance and improving the effectiveness of physical activity, as previously reported in sarcopenic aged mice [15]. Focusing on the similarities among SS-31 treatment and exercise training, the results obtained in this study confirm the partial disconnection between muscle mass and muscle strength. Indeed, the primary endpoint in most of the current clinical trials against cachexia is the maintenance or gain of lean body mass, with a minor attention to muscle function, that, however, has a relevant impact on patient quality of life. Consistently, the ROMANA trials, among the largest phase 3 clinical trials in cancer cachexia testing the effect of a ghrelin analog, showed beneficial effects of the drug in gaining lean body mass, without improving muscle strength [34]. The inclusion of SS-31 or other mitochondria-targeted compounds in a multimodal therapeutic protocol based on ghrelin analogues could improve muscle function and mass, respectively, thus more effectively benefiting cachectic cancer patients.

Wasting of voluntary skeletal muscles is considered the main hallmark of cancer cachexia; however, cardiac and respiratory muscle dysfunction have been shown to occur in parallel [35,36], potentially contributing to systemic energy wasting. A recently published paper shows that SS-31 is able to counteract cardiorespiratory muscle weakness, alleviating cardiac and respiratory myopathy in mice bearing C26 tumors [37]. The study has several differences as compared to the current one, including animal sex, that may impact on cachexia susceptibility and mitochondria profiles [38], the use of a distinct C26 clone, inducing cachexia slowly, and finally mice were not exposed to chemotherapy. The results usefully complement the current dataset showing that SS-31 targets distinct muscle compartments producing beneficial effects in tumor-bearing mice. Similarly to the present results, the study by Smuder and collaborators show that SS-31 proves unable to rescue muscle mass, however effectively improving muscle function. The mechanism proposed relies mainly on the prevention of oxidative stress, as shown in other pathological conditions such as aging sarcopenia [15] and neurotoxicity [39,40]. The use of antioxidants in the treatment of cancer cachexia is still debated and currently not considered a priority, while molecules coupling antioxidant activity and metabolic improvements are more likely to prove effective [4]. Even the antioxidant SS-31 action may theoretically be considered a double-edged sword, since beyond being beneficial for the tumor host, might potentially favor tumor growth and/or impair chemotherapy effectiveness. The current *in vitro* and *in vivo* results would seem to discard both hypotheses. The more so, considering the metabolic alterations and the energy failure induced by the tumor in host immune cells producing cancer immune escape, it is conceivable that SS-31 might sustain tumor immunity in parallel with improving muscle function, as suggested by other interventions using an immunomodulatory diet [41].

The contribution of oxidative stress to cancer cachexia can be considered as part of complex systemic alterations whose epicenter is mitochondrial dysfunction coupled to reduced oxidative metabolism. Such a hypothesis is corroborated by evidence in C26-bearing mice or in animals receiving chemotherapy, in which oxidative stress couples with impaired TCA cycle and β-oxidation, thereby increasing glucose mobilization from the liver, as long as possible [6]. Along this line, sustaining mitochondrial quality and function sound like good options in order to prevent a systemic ‘bioenergetic catastrophe’, potentially responsible for cancer patient death. In order to understand the significance of local changes and the overall effect of SS-31 administration, the current results strengthen the importance of focusing on the systemic metabolic changes imposed by tumor growth and chemotherapy (hence including the liver) rather than on skeletal muscle only. Among the more dramatic examples of metabolic cross-talk among different tissues is the SS-31-induced sparing of liver glucose and glycogen that parallels the maintenance of muscle ATP levels, providing a connection between systemic control of energy availability and muscle energetics. Along the same line, the apparently more robust SS-31 normalization effect on liver than muscle metabolome does not imply that the liver rather than the muscle is the main target of the drug. Indeed, muscle mitochondrial ‘health’ has a marked impact on the liver metabolome, as suggested by our study and in mitochondrial diseases [42]. The other way round, hepatic mitochondrial alterations have been reported in preclinical models of cachexia [43,44], along with hepatic cardiolipin dysregulation [45], suggesting that only the comprehensive analysis in distinct tissues and in the circulation allows the interpretation of this complex background.

## 4. Materials and Methods

### 4.1. Reagents

All the materials used were obtained from Merck-Sigma Aldrich (Darmstadt, Germany), unless differently specified.

### 4.2. Animals and Experimental Design

All animal experiments were cared in compliance with the Italian Ministry of Health Guidelines and the Policy on Humane Care and Use of Laboratory Animals (NRC 2011). The experimental protocol was approved by the Bioethical Committee of the University of Torino (Torino, Italy) and the Italian Ministry of Health (Aut. Nr. 579/2018-PR). Female 6 week old BALB/c mice (Charles River, Calco, Italy) were maintained on a 12:12 h dark-light cycle with controlled temperature (22 °C) and free access to food (Global Diet 2018, Mucedola, Settimo Milanese, Italy) and water during the whole experimental period. 

Tumor hosts were subcutaneously injected with 5 × 10^5^ C26 colon carcinoma cells as previously described [46]. C26 cells were maintained in vitro in DMEM supplemented with 10% FBS, 100 U/mL penicillin, 100 µg/mL streptomycin, 100 µg/mL sodium pyruvate, 2 mM L-glutamine, at 37 °C in a humidified atmosphere of 5% CO_2_ in air. The day of tumor implantation, C26 cells were detached with trypsin, resuspended in sterile saline and implanted in the back of the animals. Mouse weight and food intake were recorded every other day, starting from the day of tumor implantation, although food intake measured the average consumed by mice sharing the same cage (*n* = 6–8) and was not usable for statistical analyses. Animals were euthanized under isofluorane anesthesia at different time-points as specified below and in Appendix A. Several muscles and tissues were rapidly excised, weighed, frozen in liquid nitrogen and stored at −80 °C for further analysis. Experimental protocol A. SS-31 upon unrestricted tumor growth or chemotherapy (Appendix A). 

The animals were randomized and divided in four groups: controls (*n* = 7), SS-31 (*n* = 6) receiving daily 2 mg/kg i.p. administration of SS-31 dissolved in sterile saline, C26 colon carcinoma bearers (C26, *n* = 16) and C26 treated with chemotherapy (C26 OXFU, *n* = 16). Chemotherapy consisted in weekly, starting from day 7 of tumor growth, i.p. injection of oxaliplatin (6 mg/kg; Accord Healthcare, Milano, Italy) followed (2 hrs later) by 5-fluorouracil (50 mg/kg; Accord Healthcare) and the dose was adjusted for actual body weight. Both C26 and C26 OXFU were divided into two sub-groups (*n* = 8 each) either receiving saline or 2 mg/kg SS-31. SS-31 treatment started at day 4 or day 7 in C26 and C26 OXFU, respectively. The experiments were terminated at day 14 for C26 and at day 28 for C26 OXFU mice, before reaching the humane endpoints or animal death. Experimental protocol B. SS-31 increasing dosage in C26 OXFU mice (Appendix A).

Animals (69) were randomized and divided in three identical cohorts of 23, each subdivided into three groups, namely controls (*n* = 7), C26 OXFU (same as protocol 1; *n* = 8) and C26 OXFU receiving SS-31 (*n* = 8). SS-31 was administered at 2 mg/kg from day 7 to 13, at 5 mg/mL from day 14 to 20 and at 10 mg/mL from day 21 to 27 of tumor growth. The three animal cohorts were sacrificed at day 14, 21 or 28, respectively.

### 4.3. Mass Spectrometry Lipidomic Analysis

Intact mitochondrial enriched fractions from gastrocnemius muscle were deposited on the MALDI target with a “double layer” deposition method as follows: a 1 μL droplet of the mitochondrial enriched fractions suspension, at concentration of 0.4 mg/mL, was deposited on the MALDI target and dried under a cold air stream (first layer); the resultant solid deposition was then covered by a thin second layer (0.35 μL droplet) of the 9-AA matrix solution (20 mg/mL in 2-propanol-acetonitrile, 60:40, *v*/*v*). After solvent evaporation, the sample was analyzed. MS settings: MALDI-TOF mass spectra were acquired on a Microflex LRF mass spectrometer (Bruker Daltonics, Hamburg, Germany). The system utilizes a pulsed nitrogen laser, emitting at 337 nm; the extraction voltage was 20 kV, and gated matrix suppression was applied to prevent detector saturation. The laser fluence was kept about 10% above threshold to have a good signal-to-noise ratio. All spectra were acquired in the reflector mode using delayed pulsed extraction; spectra acquired in negative ion mode are shown in this study. Spectral mass resolutions and signal-to-noise ratios were determined by the software for the instrument, “Flex Analysis 3.3” (Bruker Daltonics). 

### 4.4. Flow Cytometry

C26 cells (40,000/cm^2^) were treated with either oxaliplatin (10 µM) or 5-fluorouracil (1 µM). H_2_O_2_ (10 µM) was used as positive control for stress-induced cell death. All the abovementioned conditions were combined or not with 2 h SS-31 (10 µM) pretreatment. 48 hours after the treatment, the cells were collected (both supernatant and trypsinized monolayer), a 50 µL aliquot was used for cell count, and the remaining part was centrifuged, washed in phosphate buffer solution (PBS) and fixed in ice-cold 70% ethanol. For cytotoxicity and cell cycle assessments, the cells were incubated at RT in PBS containing 0.4 mg/mL DNase-free RNase and 10 μg/mL propidium iodide. Dead cells were measured according to the DNA content < 2n. The experiment was performed in triplicate for each condition. Data were collected using a BD-Accuri C6 flow cytometer (BD Bioscences, Franklin Lakes, NJ, USA) and analyzed with the C6 Analysis Software (BD Bioscences) or FCS Express 4 (De Novo Software, Pasadena, CA, USA). 

### 4.5. Grasping Test

Muscle strength was assessed using a Panlab-Harvard Apparatus device (Panlab, Barcelona, Spain). Mice were positioned with both the fore- and the hind-limbs on the grid connected to a dynamometer and the force measurements were recorded after pulling the animal’s tail. Up to 10 repetitions, with 30 to 60 s of rest, were performed for each mouse and the best 3 were recorded. The analysis was performed before tumor implantation and at day 14, 21 and 28 of tumor growth, at 8 AM before animal sacrifice, depending on the experimental protocol considered. 

### 4.6. Histology and Succinate DeHydrogenase (SDH) Staining and Enzymatic Activity

During necropsy, one tibialis anterior muscle was mounted in OCT and frozen in melting isopentane for histology. Transverse sections (10 μM) were cut on a cryostat and stained for SDH incubating for 30 min at 37 °C with 1 mg/mL NTB (nitrotetrazolium blue chloride) and 27 mg/mL Na-succinate in PBS. Several pictures were taken and the whole TA section was digitally reconstructed. Fiber cross-sectional area (CSA) was determined on the whole muscle using the Image J software (freely available at https://imagej.nih.gov/ij/, accessed on 20 January 2021). The images were analyzed by three different researchers. The reliability among individuals was verified by analyzing one muscle of each group by all three researchers. The researchers were not blinded to groups when performing the analysis. As for the total SDH activity, the muscles were homogenized (5% wt/vol) in ice-cold 150 mM NaCl, 10 mM KH2PO4, 0.1 mM EGTA and centrifuged 5 min at 800 × g collecting the supernatant. 50 µL protein homogenates were incubated with 200 μL reaction buffer containing 10 mM Na-succinate, 50 μg/mL DCPIP, 10 mM phosphate buffer (pH 7.4), 2 mM KCN, 10 mM CaCl2, 0.05% BSA. The rate of absorbance decrease at 600 nm between 3 and 20 min was corrected for the protein loading and used to calculate the relative SDH activity.

### 4.7. ATP intracellular Content

Intracellular ATP concentration was determined by bioluminescence using a commercially available kit (ATP Bioluminescence Assay Kit CLS II; Roche, Basel, Switzerland). About 50 mg from gastrocnemii were homogenized in PBS (10% wt/vol). Muscle homogenates were then diluted 10 times in 100 mM Tris, 4 mM EDTA (pH 7.75), incubated 2 min at 100 °C, and centrifuged 1 minute at 1000× *g* and the supernatant collected. An aliquot of sample (50 μL) was added to 50 µL of the luciferase reagent in a multi-well black plate (96 wells – Packard, Groningen, The Netherlands). The luminescence was measured at 562 nm with an integration time of 3 s. ATP concentrations were obtained from a log-log plot of the standard curve data.

### 4.8. Western Blotting

Half gastrocnemius muscles were used for either total protein extracts or mitochondria enriched fractions. Total protein extracts were obtained by homogenization in RIPA buffer followed by centrifugation at 15,000× *g*. Mitochondria-enriched fractions were obtained by homogenization in cold mitochondrial isolation buffer 70 mM sucrose, 210 mM mannitol, 5 mM HEPES, 1 mM EGTA, 0.5% BSA (pH 7.2) followed by two centrifugation steps at 740× *g* in order to remove the insoluble fraction. Mitochondria were pelleted by two centrifugation steps at 9000× *g* and 10,000× *g* and the supernatant was collected as cytoplasmic fraction. Equal protein amounts of total protein extract (30 µg), or supernatant (30 µg) or mitochondrial protein (10 µg) were heat-denaturated in sample-loading buffer (50 mM Tris-HCl, pH 6.8, 100 mM DTT, 2% SDS, 0.1% bromophenol blue, 10% glycerol), resolved by SDS-PAGE and transferred to nitrocellulose membranes (Bio-Rad, Hercules, CA, USA). The filters were blocked with Tris-buffered saline (TBS) containing 0.05% Tween and 5% non-fat dry milk (NFDM) and then incubated overnight with antibodies (diluted in TBS containing 0.05% Tween and NFDM or BSA depending on manufacturer’s instructions) directed against BNIP3 (ab38621; Abcam, Cambridge, UK), COX IV (ab14744; Abcam), cytochrome c (556433; BD Bioscences), LC3B (L7583), PGC-1α (AB3242; Millipore, Darmstadt, Germany), PINK-1 (SAB2500794), puromycin (3RH11; Kerafast, Boston, MA, USA), SDHA (sc-377302; Santa Cruz Biotechnology). Peroxidase-conjugated IgG (Bio-Rad) were used as secondary antibodies. Membrane-bound immune complexes were detected by an enhanced chemiluminescence system (Clarity Western ECL Blotting Substrates, Bio-Rad) on a photon-sensitive film (Hyperfilm ECL). Protein loading was normalized according to GAPDH (G8795) expression or Ponceau-S staining. The quantification of the bands was performed by densitometric analysis using the TotalLab software (NonLinear Dynamics, Newcastle Upon Tyne, UK).

### 4.9. Muscle Protein Synthesis: In Vivo Surface Sensing of Translation

Muscle protein synthesis was analyzed using the surface sensing of translation (SUnSET) method [47]. 14 days after tumor transplantation, the animals, maintained in the ad libitum fed state, received an i.p. injection of 0.04 μmol puromycin per g of actual body weight dissolved in 200 μL of PBS. At exactly 25 min after puromycin administration, the mice were anesthetized, the blood was collected and the mice were euthanized by cervical dislocation. Tibialis and gastrocnemius muscles were isolated, weighed and snap-frozen in liquid nitrogen exactly 30 min after puromycin administration. Puromycin-bound nascent peptides in muscles were assessed by western blotting on total protein extracts as above described.

### 4.10. Ex-Vivo Mitochondrial Respiration Assay

Mitochondrial oxygen consumption rate (OCR) was assessed with a Seahorse XFe96 Extracellular Flux Analyzer (Agilent Technologies, Santa Clara, CA, USA). Mitochondrial-enrichment and processing protocol was adapted from manufacturer’s instructions for respiration analysis of isolated mitochondria. Briefly, mitochondria-enriched fractions were obtained from freshly extracted gastrocnemii as described in western blotting section. After the two centrifugation steps at 9000× *g* and 10,000× *g*, the pellet was resuspended in cold mitochondrial assay solution consisting of 70 mM sucrose, 220 mM mannitol, 10 mM KH_2_PO_4_, 5 mM MgCl_2_, 2 mM HEPES, 1 mM EGTA (w/o BSA; pH 7.2). After protein quantification, 0.2% BSA was added to the mitochondria and 4µg per well were plated in the Seahorse XF 96-well Cell Culture Microplate. Mitochondrial respiration analysis started in a coupling state with 10 mM glutamate/malate; 4 mM ADP and 10 mM succinate were injected sequentially to mitochondria through the Seahorse FluxPak cartridge. Non-mitochondrial respiration was measured by adding 1 µM rotenone/antimycin A. OCR values were obtained by subtracting the non-mitochondrial respiration to the ADP or succinate-coupled respiration values. The experiment was performed loading four replicates for each sample.

### 4.11. Metabolomics Analysis by Nuclear Magnetic Resonance

Plasma samples were prepared by diluting 100 μL of plasma with 500 μL of a deuterated phosphate buffer solution (pH 7.4) containing (DSS; Thermo Fisher Scientific, Waltham, MA, USA) with a final concentration of 0.5 mM to be used as a chemical shift and quantitation reference. The solution was filtered through a 10 kDa molecular weight cut-off filter (Millipore) to remove large proteins. Samples were then placed in 5-mm NMR tubes for analysis. Muscle and liver tissues for NMR analysis were prepared according to the methanol/chloroform water extraction procedure as previously described [48]. Tissue samples of about 100 mg were used and actual weights were recorded to normalize the data. NMR spectra were acquired on an Avance III 700 MHz NMR spectrometer (Bruker, Hamburg, Germany) equipped with a TXI triple resonance probe operating at 25 °C. Spectra were collected with a 1D NOESY pulse sequence covering 12 ppm. The spectra were digitized with 32,768 points during a 3.9-s acquisition time. The mixing time was set to 100 ms, and the relaxation delay between scans was set to 2.0 s. All data were finally processed using the Spectrus Processor software (version 2016.1, Advanced Chemistry Development, Toronto, ON, Canada). The spectra were zero filled to 65,536 points and apodized using a 0.3-Hz decaying exponential function, and they were fast Fourier transformed. Automated phase correction and first-order baseline correction were applied to all samples. Metabolite concentrations were quantified using the Chenomx NMR Suite (version 8.2, Chenomx, Edmonton, AB, Canada). The DSS-d6 was used as a chemical shift and quantification reference for all spectra and was set to a chemical shift of 0.00 and a concentration of 500 μM. Quantitative fitting of each spectrum was carried out in batch mode, followed by manual adjustments to correct for errors arising from spectral overlap.

### 4.12. Glycogen Assay

Liver glycogen concentration was assessed using a commercially available system (MAK016 Glycogen assay Kit, Merck-Sigma Aldrich). Briefly, liver fragments of about 50 mg were cold homogenized in water (10% w/vol) with a bead homogenizer, boiled for 5′ and centrifuged for 5′ at 13,000× *g*. The supernatant was collected and diluted 100 folds before adding 10 μL to a 96 well plate. The assay was performed following manufacturer’s instructions and using a glycogen titration curve in order to extrapolate quantitative data.

### 4.13. Statistical Analysis

All the results are expressed as means ± SD or SEM as described in the figure legends. The significance of the differences was assayed using the SPSS software (IBM, Armonk, NY, USA) and evaluated by unpaired Student’s *t*-test or ANOVA with Tukey’s post hoc test when distribution was normal (Shapiro-Wilk’s normality test) and variance was homogeneous (Levene’s test). When distribution was not normal data were analyzed with non-parametric ANOVA (Kruskal-Wallis’ test with Bonferroni’s correction). When variance was non-homogeneous, data were analyzed by heteroscedastic ANOVA with Brown-Forsythe’s procedure and Games-Howell’s post hoc test. For experimental protocol A, pair-wise comparisons through ANOVA were performed grouping control and SS-31 with C26 and C26 SS-31, or C26 OXFU and C26 OXFU SS-31. For the grip strength test and comparing day 14, 21, and 28 data (experimental protocol B), an ANOVA for repeated measures was performed, followed by Bonferroni’s correction. Differences were considered significant for *p* < 0.05. Investigators were not blinded to the experimental groups.

## 5. Conclusions

In conclusion, the results obtained targeting pharmacologically muscle mitochondria point to a defined window of effectiveness, enforcing the concept that early interventions are mandatory in order to obtain a tangible clinical benefit before refractory cachexia ensues. Overall, our data supports future therapeutic interventions aimed at testing SS-31 improved mitochondrial function as a mean to counteract energy wasting in cachexia induced by both tumor growth and chemotherapy. As fighting tumors by targeting cancer cell metabolism is becoming a reality, improving cancer patient condition targeting both tumor and host metabolism is a new challenge to improve patient’s quality of life and survival.

## Figures and Tables

**Figure 1 cancers-13-00850-f001:**
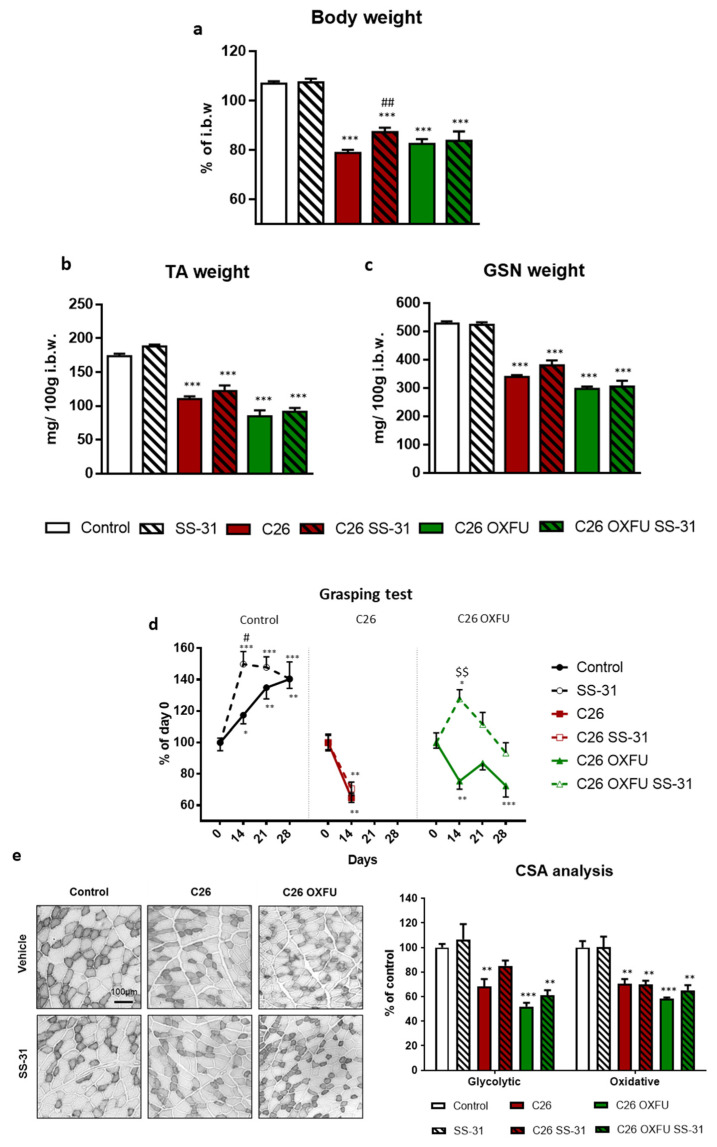
SS-31 slightly impacts on cancer- and chemotherapy-induced cachexia. Body weight (**a**), tibialis anterior weight (TA; **b**), gastrocnemius weight (GSN; **c**), grasping test (**d**) and cross sectional area (CSA; **e**; representative pictures and quantification) of glycolytic and oxidative (light and dark grey, respectively) muscle fibers of controls (*n* = 7), SS-31-treated mice (SS-31; *n* = 6), tumor-bearing mice (C26; *n* = 8), SS-31-treated tumor-bearing mice (C26 SS-31; *n* = 8) and tumor-bearing mice administered with chemotherapy alone (C26 OXFU; *n* = 8) or in combination with SS-31 (C26 OXFU SS-31; *n* = 8). Body weight data (means ± SEM) are expressed as percentage of initial body weight (i.b.w.). Muscle weight (means ± SEM) is expressed as mg of tissue per 100 g of initial body weight. Grasping test data (means ± SEM) are expressed as percentage of day 0. CSA (means ± SEM) is expressed as percentage of control. Significance of the differences: ** *p* < 0.01, *** *p* < 0.001 vs. control; ## *p* < 0.01 vs. C26. For grasping test (panel d) the significance of the difference is * *p* < 0.05, ** *p* < 0.01, *** *p* < 0.001 vs. day 0; # *p* < 0.05 vs. control; $$ *p* < 0.01 vs. C26 OXFU.

**Figure 2 cancers-13-00850-f002:**
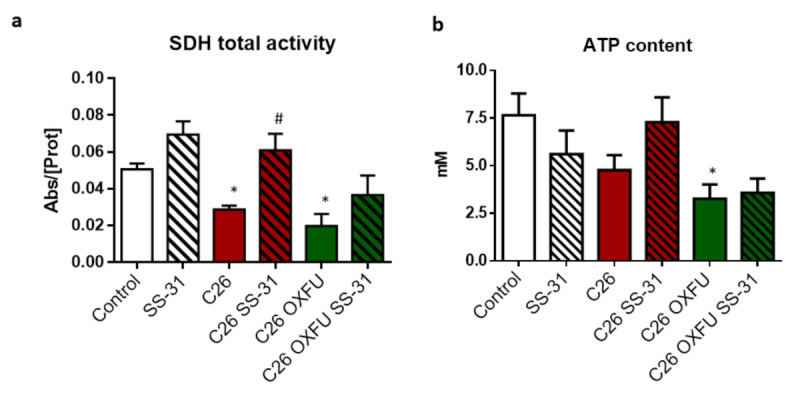
SS-31 improves mitochondrial activity without modulating protein expression. Muscle SDH total activity (**a**), ATP content (**b**) and mitochondrial marker proteins (quantification and representative blot; (**c**) of controls (*n* = 7), SS-31-treated mice (SS-31; *n* = 6), tumor-bearing mice (C26; *n* = 8), SS-31-treated tumor-bearing mice (C26 SS-31; *n* = 8) and tumor-bearing mice administered with chemotherapy alone (C26 OXFU; *n* = 8) or in combination with SS-31 (C26 OXFU SS-31; *n* = 8). SDH activity (means ± SEM) is expressed as the ratio between absorbance and total protein content. ATP content (means ± SEM) is expressed as absolute concentration (mM). Protein data (means ± SEM) are expressed as a percentage of control. Significance of the differences: * *p* < 0.05, ** *p* < 0.01 vs. control; # *p* < 0.05 vs. C26.

**Figure 3 cancers-13-00850-f003:**
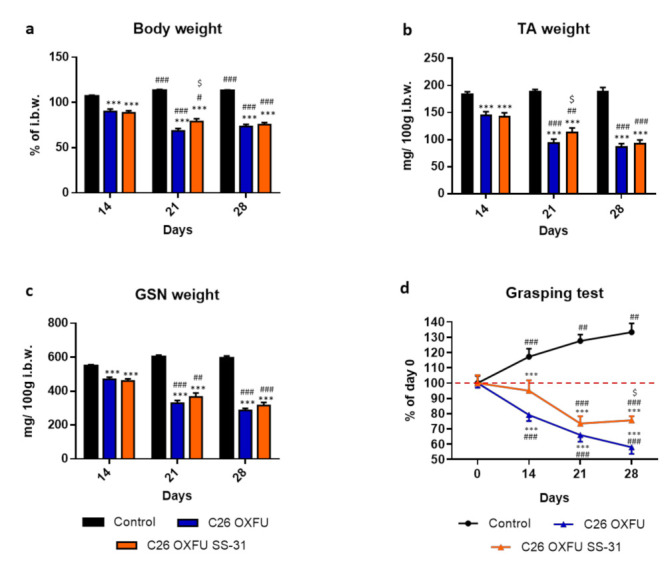
Progressively increasing SS-31 dosage show a therapeutic window in chemotherapy-treated C26-bearing mice. Body weight (**a**), tibialis anterior weight (TA; **b**), gastrocnemius weight (GSN; **c**) and grasping test of controls (*n* = 7) and tumor-bearing mice administered with chemotherapy alone (C26 OXFU; *n* = 8) or in combination with SS-31 (C26 OXFU SS-31; *n* = 8) euthanized 14, 21 or 28 days after tumor implantation. Body weight data (means ± SEM) are expressed as percentage of initial body weight (i.b.w.). Muscle weight (means ± SEM) is expressed as mg of tissue per 100g of initial body weight. Grasping test data (means ± SEM) are expressed as percentage of day 0. Significance of the differences: *** *p* < 0.001 vs. control; # *p* < 0.05, ## *p* < 0.01, ### *p* < 0.001 vs. day 14; $ *p* < 0.05 vs. C26 OXFU. For grasping test (panel **d**) the significance of the difference is *** *p* < 0.001 vs. day 0; ## *p* < 0.01, ### *p* < 0.001 vs. control; $ *p* < 0.05 vs. C26 OXFU.

**Figure 4 cancers-13-00850-f004:**
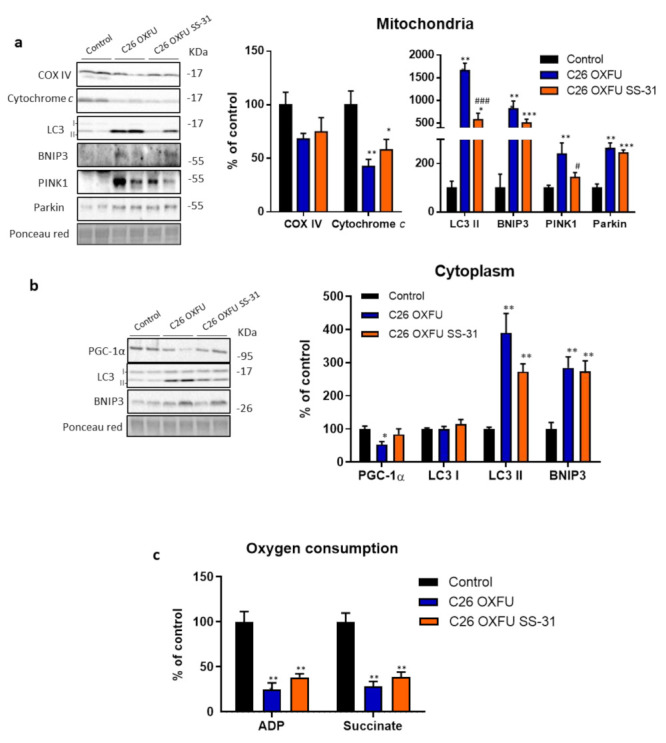
SS-31 counteracts mitochondrial loss and excessive autophagy/mitophagy. Mitochondria and autophagy/mitophagy marker proteins (representative blot and quantification) of mitochondrial (**a**) or cytoplasmic (**b**) fraction and ADP/succinate-coupled oxygen consumption (**c**) of controls (*n* = 7) and tumor-bearing mice administered with chemotherapy alone (C26 OXFU; *n* = 8) or in combination with SS-31 (C26 OXFU SS-31; *n* = 8) at day 21 after tumor implantation. Protein and oxygen consumption data (means ± SEM) are expressed as percentage of control. Significance of the difference: * *p* < 0.05, ** *p* < 0.01, *** *p* < 0.001 vs. control; # *p* < 0.05, ### *p* < 0.001 vs. C26 OXFU.

**Figure 5 cancers-13-00850-f005:**
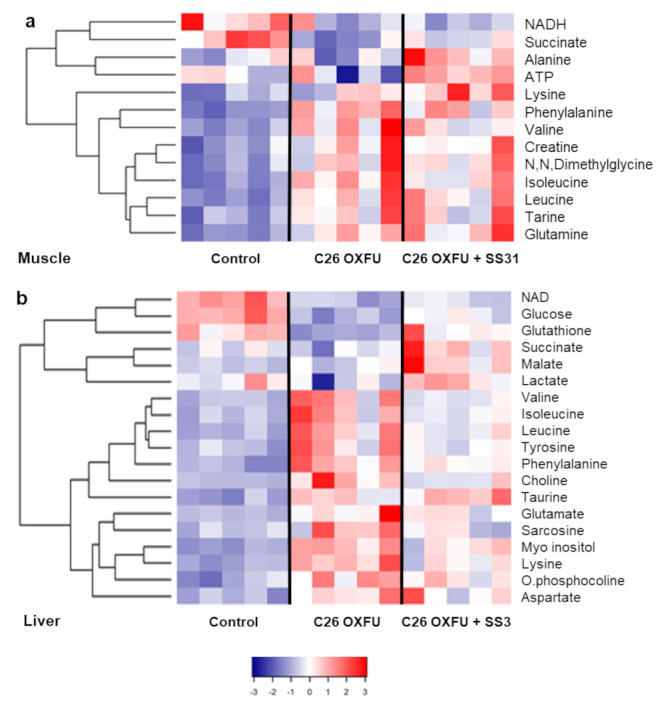
SS31 differentially impacts muscle and liver metabolome. Heatmaps of (**a**) muscle (**b**) and liver metabolites of controls (*n* = 5) and tumor-bearing mice administered with chemotherapy alone (C26 OXFU; *n* = 5) or in combination with SS-31 (C26 OXFU + SS-31; *n* = 5) at day 21 after tumor implantation. Colors represent z-scores and metabolites are presented with hierarchical cluster analysis.

## Data Availability

The data presented in this study are available on request from the corresponding author. The data are not publicly available since originated in distinct laboratories and the corresponding author is the only depositary.

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
