# Peer review of "Targeting Mitochondria by SS-31 Ameliorates the Whole Body Energy Status in Cancer- and Chemotherapy-Induced Cachexia"

_cancers, 2021, doi:10.3390/cancers13040850_

Round 1
Reviewer 1 Report
Ballarò et al. investigated the effect of mitochondria-targeted compound SS-31 administration on the change of metabolic protein expressions and mitochondrial functions using C26-implanted mice with or without chemotherapy. The authors demonstrated that SS-31 mitigated the loss of body weight and CSA of glycolytic myofibers and SDH activity in the C26 mice in the 1st experiment. In the 2nd experiment, an increased and more prolonged dose of SS-31 alleviated the loss of body and TA weights at 21 days and attenuated the muscle function loss at day 14 and 28 in the C26 OXFU mice. SS-31 mitigated the mitochondrial protein loss and aberrant autophagy activation at day 21. NMR-based metabolomics data showed that SS-31 altered the energy status in skeletal muscle and liver. The experiments were well executed and presented, and the results were promising. It can be published in the present form with a minor revision.
- Line 127. Typo. Hearth -> heart
- Line 140. The authors state, “SDH activity was maintained 139 by SS-31 … partially corrected in C26 OXFU mice.” No symbol was observed in Fig 2a on the top of the C26 OXFU SS-31 column. Were there any difference between C26 OXFU and C26 OXFU SS-31 mice? Please clarify.
- Line 463. The change of line should not be necessary here.
Author Response
We are very grateful to the reviewer for the kind evaluation of our manuscript and we apologize for the mistakes found, that have been fixed in the revised version.
Line 127. Typo. Hearth -> heart
The typo has been corrected, see line 169.
Line 140. The authors state, “SDH activity was maintained by SS-31 … partially corrected in C26 OXFU mice.” No symbol was observed in Fig 2a on the top of the C26 OXFU SS-31 column. Were there any difference between C26 OXFU and C26 OXFU SS-31 mice? Please clarify.
We thank the reviewer for pointing out this issue. Our initial aim was to state that no significant difference was found when comparing control and C26 OXFU SS-31 animals (p=0.465), as opposed to the comparison between control and C26 OXFU mice, in which the difference was statistically significant (p=0.032). However, when comparing C26 OXFU and C26 OXFU SS-31, there is no statistically significant difference (p=0.432). We modified the sentence on line 187 to clarify this point.
Line 463. The change of line should not be necessary here.
We are sorry for the lack of attention. The change of line has been removed.
Reviewer 2 Report
The paper is of interest and well written.
Perhaps a table including previous studies targeting muscle mitochondria may be of interest in the introduction part.
Author Response
We thank the reviewer for the appreciation of our work. As for the suggestion to add a table in the introduction, we modified the introduction mentioning two reviews highlighting previous attempts to improve muscle mitochondrial function in cachexia, although most of the interventions targeted mitochondria indirectly (lines 76-79). Beyond this reason, we did not add a table in the introduction also in order to avoid excessive additional weight to the manuscript that already contains 14 figures (9 supplementary).
Reviewer 3 Report
Ballarò et al. investigated the therapeutic potential of the compound SS-31 to mitigate cancer/chemotherapy-induced cachexia. The authors find SS-31 to confer some protections to muscle strength, body weight, and tibialis anterior weight in animals with cancer and chemotherapy when SS-31 is given in progressively greater doses. These protections are thought to be moderated by protections to SDH activity and a blunting of mitophagy, as well as a whole body systemic effect of the drug itself. Experimentally, the study attempts to model the clinical scenario (e.g. imitating drug treatment a few days after cancer injection and the inclusion of chemotherapy treatment) which facilitates a more “realistic” interpretation of the viability of SS-31 and cachexia overall. Overall, the study is well designed and well written; however I have a few comments/suggestions/questions listed below:
Abstract:
A more specific Purpose statement would be a nice addition.
I don’t think “exercise mimetic” is appropriate wording here, especially since you don't really mention it/describe it in the introduction.
Introduction
Mitochondrial alterations as a possible mechanism is an important topic for investigation, though I would suggest the inclusion of Brown et al. JCSM, 2017. Using a similar model (lewis lung carcinoma) the authors find mitochondrial alterations precede the development of muscle loss, which would provide more rationale for the current study.
Methods:
Line 391: This is a pretty young age for mice, is there a specific reason the authors chose 6 weeks instead of an older time point?
Line 394: It may be helpful to provide a sentence or two of why the dosage was selected since readers new to the cachexia literature may be unfamiliar with this model.
Line 414: Why were these experiments terminated at different times? It's probably somewhere else in the manuscript, but adding it here would be helpful as well.
Line 421: was n=8 in each of these cohorts? Or was 8 divided across the three cohorts?
Line 451-457: would be nice to mention how much rest was allowed in between each repetition since rest time can influence maximal grip strength. Also, was grip strength always performed the same time of day?
Line 461: Can the authors more fully describe how CSA by SDH phenotype was quantified? Was it the same person for all images? Or multiple individuals? If multiple individuals was there validation of reliability between individuals? Were researchers blinded to groups when performing this analysis?
Line 505: Where animals in a fasted or fed state for puromycin injections?
For some of these protocols it seems like n=8 for 6 groups may be slightly underpowered for a full model ANOVA, and it seems like in a full model ANOVA it may be making pair-wise comparisons that are not informative for the primary research question. It may be more helpful for the authors to "condense the model" to the primary comparisons of interest and simply state those in the statistical section. Model condensing can be done using the LSMestimate function or contrast statements in SAS or if the authors' prefer, I would find it acceptable to pre-state comparisons of interest, simply perform t-test sand then adjust for number of comparisons bonferroni or false-discovery rate, or depending on number of comparisons accepting a slightly inflated Type I error rate and report exact p-values.
Results
I like the experimental design figures.
Supplementary Figure 3: Since SF are not typically printed, perhaps using color in this figure would help better distinguish between groups.
Please give some idea of magnitude of effect for differences (e.g. Body weight was 30% lower in X group compared to X group).
Line 116: think this should be “consistent”
I may have missed this, but did all the C26 die before 21 days?
Figure 1: quick caution on interpretation, just because a group (e.g. c26 +ss-31) is not different from another group does not necessarily mean they are equivalent, testing equivalency is a separate statistical test. My guess is, due to an ANOVA with 6 groups, that may have squashed significance.
Quick note, "increase/decrease" imply longitudinal data. Although there are some longitudinal outcomes, technically tissue weights are cross sectional, so it would be more appropriate to use "greater" / "less than"
Line 137: I think this is in the methods but probably nice to remind the reader which muscle.
Line 141: "trend analysis" is actually a specific type of analysis. I would suggest something more like "approached significance" and denote exact p value, that way readers can have their own interpretation of these findings.
Curious why the authors used Ponceau as an internal control for some images and gapdh in other sections?
Given that BNIP3 dimerizes, in my experience this typically results in a much longer band (typically a bandaround 50kda and 30ish kDa) than what I'm seeing here. This may require taking another look and possibly reanalyzing the blots for both BNIP3 bands.
Is it possible to increase the text in Fig 5?
Discussion
line 254, I would soften this language, there's a lot that goes into developing clinical diagnostic tests and there's quite a few attempts to determine cachexia diagnosis.
Line 256: Is energy failure the correct word here? I imagine to warrant that statement, I would want to see more energy expenditure specific data. Also regardless of SS-31 they still appeared to lose muscle mass and fat, so it seems a bit strong to suggest that SS-31 counteracted "energy failure".
Line 276: I think it would be helpful to refer the reader to the data that showed this, since there is quite a bit.
Do the authors have any hypothesis why SS-31 itself did not appear to have any of these effects/mitochondrial protections in the CON-SS-31 mice?
Line 286: To make this argument, I would like to see measures of mitochondrial dynamics. So I would suggest, add a stipulation here.
Line 290-293: these data may be worth showing in a supplementary figure.
Line 312: I would caution here, SS-31 itself didn't appear to show adaptations similar to exercise compared to control (e.g. greater SDH+ fibers, PGC content, COXIV content etc.). I think it's fine to say some of the effects it may confer are similar to exercise; but considering the data "exercise mimic" seems a bit overreaching.
Line 326: it seems based on the presented data, SS-31 may have some benefit (at least early) with muscle function, it seems less conclusive for the support maintenance of muscle mass.
Line 329: semi-colon
Line 332-336: That's a pretty long sentence, perhaps break up a little bit.
It may be worth noting quickly in the discussion that this study was completed in female mice and there is evidence to suggest males and females have different mitochondrial profiles (see Rosa-Caldwell & Greene, J. Biol Sex Diff, 2019) and cachexia susceptibility (see Hetzler et al. Biochim Biophys Acta, 2015), so it is possible the sex of the animals may have influenced the potency of the intervention.
Author Response
We are very grateful to the reviewer for nicely summarizing the major findings of our work and placing them in the right context. We thank for the constructive criticisms that helped us improving the manuscript. A point-by-point reply to the points raised follows.
Abstract:
A more specific Purpose statement would be a nice addition.
The aim of the study has been rephrased in order to mention the focus on metabolic alterations (lines 31-32). The whole abstract has been revised to reduce it according to the journal’s guidelines and the Simple Summary was added.
I don’t think “exercise mimetic” is appropriate wording here, especially since you don't really mention it/describe it in the introduction.
The sentence including “exercise mimetic” has been condensed, removing such definition (line 31).
Introduction
Mitochondrial alterations as a possible mechanism is an important topic for investigation, though I would suggest the inclusion of Brown et al. JCSM, 2017. Using a similar model (lewis lung carcinoma) the authors find mitochondrial alterations precede the development of muscle loss, which would provide more rationale for the current study.
We welcome the suggestion. The mentioned paper has been briefly described and included in the introduction (lines 74-76)
Methods:
Line 391: This is a pretty young age for mice, is there a specific reason the authors chose 6 weeks instead of an older time point?
We agree that this a pretty young age and that in humans cancer occurs mainly above middle age. However, in our experience, the most reliable data with the C26 model are obtained in young animals, in terms of tumor size, cachexia susceptibility and data dispersion. Finally as discussed below, young mice are less affected by sex hormones than adult mice, thus making less relevant the choice of a given gender.
Line 394: It may be helpful to provide a sentence or two of why the dosage was selected since readers new to the cachexia literature may be unfamiliar with this model.
The C26 model is widely used and is the most common cachexia model with about 700 references; however, we added a reference to a paper in which we extensively described the model and the procedures required for reproducing the model (line 504).
Line 414: Why were these experiments terminated at different times? It's probably somewhere else in the manuscript, but adding it here would be helpful as well.
The distinct endpoints were chosen based on animal survival. Without chemotherapy, C26-bearing mice start dying at 14 days of tumor growth, while chemotherapy treated ones survive for much longer, in our experimental conditions and using this cell clone. We have clarified the point in the text, see line 524.
Line 421: was n=8 in each of these cohorts? Or was 8 divided across the three cohorts?
We apologize for the potential misunderstanding. We changed the sentence to: “69 animals were randomized and divided in three identical cohorts of 23, each subdivided in 3 groups, namely controls (n=7), C26 OXFU (same as protocol 1; n=8) and C26 OXFU receiving SS-31 (n=8)”. See lines 526-528.
Line 451-457: would be nice to mention how much rest was allowed in between each repetition since rest time can influence maximal grip strength. Also, was grip strength always performed the same time of day?
The grasping test was always performed at 8 AM before animal sacrifice. The resting time is between 30 and 60 seconds, required to assay the 7-8 mice per cage before testing again the first one. The text now include such information (lines 562-566).
Line 461: Can the authors more fully describe how CSA by SDH phenotype was quantified? Was it the same person for all images? Or multiple individuals? If multiple individuals was there validation of reliability between individuals? Were researchers blinded to groups when performing this analysis?
SDH stained section were analyzed by measuring the dark grey fiber (oxidative) and the light grey fiber (glycolytic) cross sectional area with ImageJ. Several pictures were taken with 10X phase contrast microscope objective and then the whole TA section was digitally reconstructed. Due to the high volume of pictures, these were analyzed by 3 different researchers balancing the number of sections to be analyzed per experimental group. The reliability among individuals was verified by analyzing one muscle of each group by all the 3 researchers. Thus, beyond the assigned sections, all of us analyzed the same 4 animals belonging to the different 4 groups. The difference between our analyses was below 10%, so we considered them reliable. The researchers were not blinded to groups when performing the analysis. Such methodological details have been added (lines 571-576)
Line 505: Where animals in a fasted or fed state for puromycin injections?
The animals were in a fed state. For ethical reasons, we are not allowed to fast cachectic tumor-bearing animals. The fed state has been clarified in the text (line 619).
For some of these protocols it seems like n=8 for 6 groups may be slightly underpowered for a full model ANOVA, and it seems like in a full model ANOVA it may be making pair-wise comparisons that are not informative for the primary research question. It may be more helpful for the authors to "condense the model" to the primary comparisons of interest and simply state those in the statistical section. Model condensing can be done using the LSMestimate function or contrast statements in SAS or if the authors' prefer, I would find it acceptable to pre-state comparisons of interest, simply perform t-test sand then adjust for number of comparisons bonferroni or false-discovery rate, or depending on number of comparisons accepting a slightly inflated Type I error rate and report exact p-values.
We definitively agree with this observation. Indeed, for our analysis, we performed an ANOVA grouping the cohorts of animals in accordance with our research question. For the experimental protocol A, instead of performing an ANOVA on all the groups at the same time, we condensed our analysis performing a first ANOVA grouping controls, SS-31, C26, C26 SS-31 and a second analysis grouping controls, SS-31, C26 OXFU, C26 OXFU SS-31, both analysis followed by Tukey’s post hoc test. In this way, we were able to avoid useless pair-wise comparison and control the type I error adequately. We are sorry for the lack of attention in stating this in the statistical section. A sentence was added, see lines 683-685.
Results
I like the experimental design figures.
We thank for the positive feedback. We hope the schematic representation help the reader understand the experimental conditions adopted. It is frequent to read about the use of C26-bearing mice, although there might be relevant different uses of the same tumorigenic cells, making the results hard to interpret. Moreover, having sacrificed the animals at distinct time points, we believe it is fundamental to depict the experimental design.
Supplementary Figure 3: Since SF are not typically printed, perhaps using color in this figure would help better distinguish between groups.
Figure S3 has been replaced adding colors, using the same for each cytotoxic compound.
Please give some idea of magnitude of effect for differences (e.g. Body weight was 30% lower in X group compared to X group).
The results on body and tissue weight are now described including the % of the differences between the groups (line 135 and beyond).
Line 116: think this should be “consistent”
We apologize for the mistake and changed “consistently” to “consistent”.
I may have missed this, but did all the C26 die before 21 days?
The animals bearing the C26 tumor, not receiving chemotherapy, were sacrificed 14 days after tumor injection, as explained in the methods and in the experimental design (Figure S2a). In our experience, those animals start dying at 14 days and most are dead within one further week. However, in the current study, no survival experiment was performed.
Figure 1: quick caution on interpretation, just because a group (e.g. c26 +ss-31) is not different from another group does not necessarily mean they are equivalent, testing equivalency is a separate statistical test. My guess is, due to an ANOVA with 6 groups, that may have squashed significance.
We totally agree with the reviewer ant thank for the valuable comment. Beyond the comment on grouping the animal cohorts for the stats (see above), we believe that little changes, even if statistically demonstrated, have potentially little translational significance. The other way round, when moving to more chronic conditions (i.e. tumor + chemotherapy) and increasing SS-31 dosage, we succeeded in demonstrating statistically significant differences even for body and muscle weight. Of note, the current study suggests that targeting mitochondria impacts on tissue function rather than on tissue mass. The paramountcy of muscle and other tissue mass vs function in cancer patient survival and quality of life is a matter of debate in the last years.
Quick note, "increase/decrease" imply longitudinal data. Although there are some longitudinal outcomes, technically tissue weights are cross sectional, so it would be more appropriate to use "greater" / "less than"
The text has been accordingly modified for the description of body and tissue weight related to both Figure 1 and 3, starting from line 136.
Line 137: I think this is in the methods but probably nice to remind the reader which muscle.
We apologize for the lack of clarity. Now the use of gastrocnemius muscle has been specified.
Line 141: "trend analysis" is actually a specific type of analysis. I would suggest something more like "approached significance" and denote exact p value, that way readers can have their own interpretation of these findings.
Reviewer 1 also noticed our mistake. Here follows the response: “We thank the reviewer for pointing out this issue. Our initial aim was to state that no significant difference was found when comparing control and C26 OXFU SS-31 animals (p=0.465), as opposed to the comparison between control and C26 OXFU mice, in which the difference was statistically significant (p=0.032). However, when comparing C26 OXFU and C26 OXFU SS-31, there is no statistically significant difference (p=0.432). We modified the sentence on line 187 to clarify this point.”
Curious why the authors used Ponceau as an internal control for some images and gapdh in other sections?
Regarding the western blotting of the experimental protocol B, in which we separated cytosolic and enriched mitochondrial fractions, we experienced issues in founding a reliable reference protein for normalizing the expression of the protein of interest in the mitochondrial fraction. For this reason, we used Ponceau staining for normalizing and we decided to also adopt it, for consistency of the analysis, to the cytosolic fraction.
Given that BNIP3 dimerizes, in my experience this typically results in a much longer band (typically a band around 50kda and 30ish kDa) than what I'm seeing here. This may require taking another look and possibly reanalyzing the blots for both BNIP3 bands.
We did several attempts to reveal the BNIP3 band in mitochondria and we were not able to see any monomeric 30 kDa or around band. The only bands that appeared were at around 55 and over 72 kDa, likely the 55 kDa band representing the BNIP3 dimer. An explanation of this result may rely on the fact that BNIP3 on mitochondria manly exists as dimeric form (PMID: 19641497). Regarding the cytosolic fraction, we did find many bands in our blot at around 55KDa in addition to the band found at around 26KDa. Since the BNIP3 band as dimer was represented in the mitochondria fraction in our experiment, we analyzed only the monomeric form in the cytoplasm (corresponding to the one that is more frequently reported in published papers), representing the available source of the protein for dimerization and activation on the mitochondria.
Is it possible to increase the text in Fig 5?
We apologize for the low readability. Both Figure 5 and Figure S9 have been revised in order to improve the ability to read the metabolites on the right legend.
Discussion
line 254, I would soften this language, there's a lot that goes into developing clinical diagnostic tests and there's quite a few attempts to determine cachexia diagnosis.
The sentence has been down-toned, trying to keep the concept that energy metabolism alterations may provide a useful target for diagnosing and following cachexia evolution. The revised sentence is: “ Energy metabolism impairment can be easily detected non-invasively and may potentially allow to monitor the onset and progression of cachexia from a systemic point of view ”.
Line 256: Is energy failure the correct word here? I imagine to warrant that statement, I would want to see more energy expenditure specific data. Also regardless of SS-31 they still appeared to lose muscle mass and fat, so it seems a bit strong to suggest that SS-31 counteracted "energy failure".
Also this sentence has been modified keeping more close to the results and reducing the speculation. The revised sentence is: “The results reported in the present study push on mitochondrial efficiency as a tool to improve muscle and systemic energy metabolism.”
Line 276: I think it would be helpful to refer the reader to the data that showed this, since there is quite a bit.
Now the text refers to the results shown in Figure S1.
Do the authors have any hypothesis why SS-31 itself did not appear to have any of these effects/mitochondrial protections in the CON-SS-31 mice?
SS-31 administration to healthy control mice produced a transient strength increase (Figure 1) and improved muscle SDH activity (Figure 2), with no or little improvement of muscle and fat mass (Figure 1 and S4, respectively). The improved SDH activity is in line with the mito-protective SS-31 action, as discussed (see lines 480-482). The other way round, being SS-31, at least in the current treatment conditions, unable to modulate mitochondrial abundance, we did not expect major muscle changes to happen, especially in healthy animals.
Line 286: To make this argument, I would like to see measures of mitochondrial dynamics. So I would suggest, add a stipulation here.
We agree with the reviewer. In the current work, mitochondrial dynamics were not considered since in a recent paper (see Ref 7 in the manuscript, PMID 31150737) we showed that in the muscle of C26-bearing mice, the genes controlling mitochondrial dynamics (including the fission-related DRP1) are repressed despite increased autophagy/mitophagy and impaired mitochondrial respiration. Moreover, our treatments lasted a maximum of 3 weeks, while in the cited paper on heart improvement, SS-31 was used for 3 months to restore mitochondrial dynamics. We have added a sentence in the manuscript- mentioning that the speculation on mitochondrial damage would require further investigation (lines 394-395).
Line 290-293: these data may be worth showing in a supplementary figure.
Unfortunately, the mentioned data were obtained in distinct plates and not homogeneously in a single assay to allow a correct statistical elaboration and presentation. We rely on the reviewer’s suggestion to either leave or remove the sentence.
Line 312: I would caution here, SS-31 itself didn't appear to show adaptations similar to exercise compared to control (e.g. greater SDH+ fibers, PGC content, COXIV content etc.). I think it's fine to say some of the effects it may confer are similar to exercise; but considering the data "exercise mimic" seems a bit overreaching.
We agree that, whenever possible, exercise is the best and stronger intervention. SS-31 only mimics some of the beneficial actions exerted by exercise. Indeed, in the mentioned sentence, we propose SS-31 as a plus to improve exercise toreance/effectiveness, not as a replacement. The sentence is now revised avoiding the use of “exercise mimetic” (line 422).
Line 326: it seems based on the presented data, SS-31 may have some benefit (at least early) with muscle function, it seems less conclusive for the support maintenance of muscle mass.
We apologize for the misunderstanding. The idea was that ghrelin can support muscle mass, while SS-31 or other mitochondria-targeted compounds may sustain muscle function. The sentence has been rephrased in order to avoid potential misunderstandings and now is: “The inclusion of SS-31 or other mitochondria-targeted compounds in a multimodal therapeutic protocol based on ghrelin analogues could improve muscle function and mass, respectively, thus more effectively benefiting cachectic cancer patients”.
Line 329: semi-colon
The comma was changed with a semi-colon.
Line 332-336: That's a pretty long sentence, perhaps break up a little bit.
The period and the previous one have been reformulated, see lines 438-443.
It may be worth noting quickly in the discussion that this study was completed in female mice and there is evidence to suggest males and females have different mitochondrial profiles (see Rosa-Caldwell & Greene, J. Biol Sex Diff, 2019) and cachexia susceptibility (see Hetzler et al. Biochim Biophys Acta, 2015), so it is possible the sex of the animals may have influenced the potency of the intervention.
This is a really valuable comment. When we designed the experiment, we decided, based on extensive previous experience with both male and female mice bearing C26 tumors, to use females in order to reduce fighting and cannibalism in frail animals that we frequently observed when using males. Moreover, the age chosen is pretty low, potentially being an issue (as previously discussed) for the translatability to humans in term of cancers mainly occurring in middle-aged to aged individuals, however limiting the impact of sex hormones on the results. Indeed, in young mice, C26 tumors induce cachexia very similarly in males and females (see PMID: 26343953). Consistently, the study by Smuder (see ref 39, PMID: 33014286), performed on male mice bearing C26 tumors and treated with SS-31, produced results in line with our observations. However, the results might differ in aged animals and especially in humans, thus we mentioned this relevant aspect that was not previously discussed (lines 440-443).